# Oxygen Vacancy and Interface Effect Adjusted Hollow Dodecahedrons for Efficient Oxygen Evolution Reaction

**DOI:** 10.3390/molecules28155620

**Published:** 2023-07-25

**Authors:** Huan Wang, Qian Ma, Fengmin Sun, Yachuan Shao, Di Zhang, Huilan Sun, Zhaojin Li, Qiujun Wang, Jian Qi, Bo Wang

**Affiliations:** 1Hebei Key Laboratory of Flexible Functionals Materials, School of Materials Science and Engineering, Hebei University of Science and Technology, Shijiazhuang 050000, China; wanghuantp@163.com (H.W.); m15612152902@163.com (Q.M.); sfmkydn@126.com (F.S.); shaoyachuan1998@163.com (Y.S.); snowfoxzd@hebust.edu.cn (D.Z.); sunhuilan1949@163.com (H.S.); lizhaojin1214@163.com (Z.L.); wangqiujun090@126.com (Q.W.); 2State Key Laboratory of Biochemical Engineering, Institute of Process Engineering, Chinese Academy of Sciences, Beijing 100049, China

**Keywords:** metal–organic frameworks, cobalt–cerium composite oxide, phase interface, hollow dodecahedrons, oxygen evolution reaction

## Abstract

Metal–organic frameworks (MOFs) with special morphologies provide the geometric morphology and composition basis for the construction of platforms with excellent catalytic activity. In this work, cobalt–cerium composite oxide hollow dodecahedrons (Co/Cex-COHDs) with controllable morphology and tunable composition are successfully prepared via a high-temperature pyrolysis strategy using Co/Ce-MOFs as self-sacrificial templates. The construction of the hollow structure can expose a larger surface area to provide abundant active sites and pores to facilitate the diffusion of substances. The formation and optimization of phase interface between Co_3_O_4_ and CeO_2_ regulate the electronic structure of the catalytic site and form a fast channel favorable to electron transport, thereby enhancing the electrocatalytic oxygen evolution activity. Based on the above advantages, the optimized Co/Ce0.2-COHDs obtained an enhanced oxygen evolution reaction (OER) performance.

## 1. Introduction

The four-proton-coupled electron mechanism in the electrocatalytic oxygen evolution reaction (OER) is the main reason for its sluggish kinetics and high overpotential, which severely restrict the efficiency of related energy storage and conversion technologies [1,2,3]. Designing and developing high-performance electrocatalysts for oxygen evolution that promote O-H bond cleavage and O-O bond formation is of great significance and poses a significant challenge [4,5,6]. Currently, Ru, Ir and their oxides are the only commercial catalysts, and their large-scale applications are severely limited by high cost. Therefore, research on efficient, inexpensive, and stable oxygen evolution electrocatalyst is imminently required [7,8,9].

In the family of transition metal materials, Co_3_O_4_ has been favored by many researchers due to its variable valence, flexible and tunable structure, and high stability [10,11,12,13]. Constructing composite structures can optimize the composition and microstructure (interfaces, grain size, defects, etc.) of the material to regulate the electronic behavior, which will contribute to the enhancement of the catalytic activity [14,15,16,17,18]. In addition to Co_3_O_4_ as a matrix, the selection of the second component is crucial. CeO_2_, with its strong ability to shift between Ce^3+^ and Ce^4+^ derived from the unique 4f5d electronic configuration of Ce, exhibits excellent properties including reversible surface oxygen exchange, high oxygen storage/release capacity, and good ionic/electronic conductivity [19,20,21,22]. Thus, the construction of Co_3_O_4_ and CeO_2_ into a composite structure (Figure 1) can generate abundant phase interfaces for fast charge transfer, optimize the geometric morphology, and regulate the defect distribution and electronic structure [23,24,25]. Moreover, assembling such composite nanoparticles with optimized geometry and electronic behavior to form a specific 3D micro-nano structure can avoid the loss of active sites, confine and enrich reactants, and accelerate charge transfer and ion diffusion (Figure 1). As a unique microstructure with tunable composition and morphology parameters, the hollow dodecahedron structure is widely used in many fields such as catalysis, batteries, supercapacitor, optoelectronics, luminescence, and sensors [26,27,28,29].

In this work, a series of cobalt–cerium composite oxide hollow dodecahedrons (CoCex-COHDs) with tunable Ce ratios (x) were successfully prepared by a hard-template strategy using MOFs as templates (Figure 1a and Appendix A). The modulation of Ce ratio endows composite oxide nanoparticles with reduced grain size for increasing active sites, abundant heterointerfaces as fast charge transport channels, and optimized defects and electronic structures to promote intrinsic activity. The successful construction of composite nanoparticles in the space utilization and effective surface area of the materials enhances the confinement of reactant molecules, accelerates ion transport and gas diffusion, and further improves oxygen evolution kinetics. Based on these advantages, the optimized Co/Ce0.2-COHDs achieves a low overpotential of 316 mV at 10 mA·cm^−2^, and its Tafel slope (89.9 mV·dec^−1^) is also smaller than that of other CHDs.

## 2. Results and Discussion

A typical process for preparing Co/Cex-COHDs is displayed in Figure 2. First, the metal salts and 2-methylimidazole were separately dissolved in methanol. Then, the Co/Ce-MOFs were obtained after mixing the two solutions and aging. These MOFs were further characterized by scanning electron microscope (SEM).

Figure 1a and Appendix A present SEM images of MOFs with a Ce ratio of 0.2. It can be seen from these SEM images that the MOF templates exhibit a dodecahedral structure with a smooth surface and the average size is around 1.5 μm. The MOFs mainly contain O, Co, and Ce elements, a situation which is supported by the SEM-Mapping images (Appendix A). These special morphological and structural features are the premise for the formation of the hollow dodecahedron oxide. Subsequently, Co/Ce0.2-COHDs can be prepared by pyrolyzing the MOF in air to remove elements such as C and H. As shown in Figure 1b, Co/Ce0.2-COHDs also retain the dodecahedral shape derived from the pyrolysis of MOFs. Meanwhile, the average size of Co/Ce0.2-COHDs is only about 500 nm, which is significantly smaller than that of the MOFs. This size change is mainly due to volume shrinkage during high-temperature pyrolysis. However, the surface of Co/Ce0.2-COHDs is relatively rough, and it can also be clearly seen from the SEM image (Figure 1b) that it is assembled from a large number of nanocrystals. Subsequently, the effect of Ce ratio on the microstructure of the samples was investigated by SEM characterization. The SEM images of Co/Ce0.2-COHDs with other Ce ratios (x = 0, 0.1, and 0.3) ratios are also shown in Figure 1c–e, respectively. The change in the sample morphology with the Co/Ce ratio can be clearly observed. When the Ce ratio is less or equal to 0.2, Co/Cex-COHDs can maintain the dodecahedron structure. As the Ce ratio continues to increase (x ≥ 0.3), the microstructure of Co/Cex-COHDs will collapse and cannot maintain its structural shape. In addition, the introduction of Ce will lead to smaller grains and pores, which will be beneficial to the increase in specific surface area. Furthermore, the influence of calcination temperature on Co/Cex-COHDs was also investigated. It can be seen from the SEM images in Appendix A that the calcination temperature has little effect on the sample shape. However, the higher calcination temperature will lead to larger grain size and more surface pores. Further, the internal structure of Co/Ce0.2-COHDs was observed by a transmission electron microscope (TEM). In Figure 1f, the interior of Co/Ce0.2-COHDs is hollow, and a core remains inside. This structure will bring larger effective specific surface area and higher space utilization [30]. Then, N_2_ adsorption–desorption tests were performed on the materials to reveal their specific surface area and pore distribution (Appendix A). The Co/Ce0.2-COHDs exhibit a larger specific surface area of 45.8 m^2^g^−1^ than Co_3_O_4_-HDs (31.6 m^2^g^−1^), which can be calculated from the N_2_ adsorption–desorption isotherms (Appendix A). As can be seen from the pore distribution curves (Appendix A), the pore size distribution and pore volume of Co/Ce0.2-COHDs are also larger.

The distribution of elements on the entire microstructure will affect the phase composition and catalytic activity. For this, elemental mappings were obtained by TEM testing. As shown in Figure 2a, the three elements Co, Ce, and O are uniformly distributed throughout the dodecahedral structure without any segregation, implying that the proportion of each phase in the material is also uniform. The uniform distribution of elements allows the formation of the two oxide materials with more nano–heterogeneous interfaces, which is more conducive to electron transfer. The crystallinity and phase composition of the catalyst can be analyzed by high-resolution transmission electron microscope (HRTEM) and selected area electron diffraction (SAED) images. The high-resolution transmission electron microscope (HRTEM) image in Figure 2b shows clear lattice fringes of Co/Ce0.2-COHDs, indicating its good crystallinity, and two lattice fringe spacings of 3.12 Å and 2.85 Å correspond to the (111) plane of CeO_2_ and the (220) plane of Co_3_O_4_ [31,32]. It was also observed that their lattice edges extended to each other and interlaced with each other at the interface, which indicated that CeO_2_ nanoparticles had been anchored to Co_3_O_4_ nanoparticles and formed Ce/Co composite oxide. The SAED image (Figure 2c) presents clear and bright rings, which can be indexed to the (111) and (220) planes of CeO_2_, and the (220), (311), and (422) planes of Co_3_O_4_. These results further validate the successful formation of CeO_2_ and Co_3_O_4_ composite.

The phase composition of Co/Cex-COHDs was further studied by X-ray powder diffraction (XRD). The XRD patterns of Co/Cex-COHDs with different Ce ratios are shown in Figure 2d. In XRD patterns (Figure 2d), the diffraction peaks at 31.3°, 36.8°, 38.6°, 44.8°, 55.7°, 59.5°, and 65.2° correspond to the (220), (311), (222), (440), (422), (511), and (440) planes of Co_3_O_4_, and the diffraction peaks at 28.5°, 47.5°, 56.0°, and 59.1° are attributed to (111), (220), (311), and (222) of CeO_2_, respectively [21]. The XRD patterns show that the diffraction peak intensity of CeO_2_ gradually increases while that of Co_3_O_4_ gradually decreases with the decrease in Ce ratio. As is well known, the Scherrer equation can establish the corresponding relationship between FWHW and grain size. The FWHW of the samples can be obtained through Jade software. Meanwhile, Appendix A shows the trend of the grain size changes with Ce introduction in these samples. It is also worth noting that the decrease in Ce ratio causes the full width at half maximum (FWHW) of the diffraction peaks to gradually increase, which is due to the smaller grain size of the two phases resulting from the introduction of Ce (Appendix A). Generally, Co_3_O_4_ crystallizes and grain grows faster and can easily form larger grains. The crystallization and grain growth of CeO_2_ is slow, making it easy to form smaller grains. Thus, the combination of CeO_2_ and Co_3_O_4_ can restrain each other and cause the grain size of the two to become smaller. The formation of smaller grains leads to an increase in specific surface area, more exposed active sites, and an increase in the number of pores, which ultimately leads to an increase in catalytic activity for oxygen evolution. In addition, XRD pattern (Appendix A) shows that the phase composition of the sample did not change with the increase in calcination temperature. However, it also can be clearly seen that the crystallinity of the sample is obviously better with the increase in calcination temperature, which indicates the increase in grain size.

X-ray photoelectron spectroscopy (XPS) was used to investigate and analyze the elemental composition, elemental chemical state, and electron transfer of the prepared materials. Firstly, the XPS survey spectrum of Co/Ce0.2-COHDs (Figure 3a and Appendix A) clearly shows the presence of Ce, Co, and O elements, which is consistent with the result of elemental mapping. The XPS survey spectrum (Figure 3a) of Co_3_O_4_-HDs confirms that it is composed of Co and O. The high-resolution Ce 3d XPS spectrum consists of two spin-orbiting binaries and two oscillating satellites (Figure 3b). The adsorption peaks located at 916.41 eV and 898.05 eV are attributed to Ce^4+^, while the adsorption peaks at 902.30 eV and 883.99 eV correspond to Ce^3+^. The coexistence of Ce^4+^ and Ce^3+^ provides the potential for electron transfer and electronic coupling at the two-phase interface. For this reason, the high-resolution Co 2p XPS spectra of Co_3_O_4_-HDs and Co/Ce0.2-COHDs were examined to investigate the electron transfer and change in the valence states of Co due to Ce introduction. In Co 2p XPS spectrum of Co/Ce0.2-COHDs (Figure 3c), the absorption peaks with binding energies of 796.56 eV and 781.17 eV are assigned to Co^2+^ 2p_1/2_ and Co^2+^ 2p_3/2_, and the adsorption peaks with binding energies of 794.84 eV and 779.68 eV are retrieved as Co^3+^ 2p_1/2_ and Co^3+^ 2p_3/2_, respectively. The Co 2p XPS spectrum of Co_3_O_4_-HDs (Figure 3d) shows that the adsorption peaks of Co^2+^ 2p_1/2_, Co^2+^ 2p_3/2_, Co^3+^ 2p_1/2_, and Co^3+^ 2p_3/2_ are located at 796.81 eV, 781.37 eV, 795.02 eV, and 779.81 eV, respectively. Comparing the two XPS spectra, it can be found that the Co 2p XPS spectrum of Co/Ce0.2-COHDs is shifted to the lower binding energy direction relative to the spectrum of Co_3_O_4_-HDs, suggesting that the electron transfer occurs at the two-phase interface of Co_3_O_4_ and CeO_2_. Then, the XPS was further used to elucidate the effect of Ce introduction on Co valence. After fitting and calculating the peaks of the XPS spectrum, it was found that the ratio of Co^2+^/(Co^3+^ + Co^2+^) in Co_3_O_4_-HDs is only 38.6%, while that in Co/Ce0.2-COHDs is as high as 42.3%. Such a change demonstrates that the introduction of Ce significantly increases the content of Co^2+^ [32,33]. Meanwhile, the increase in Co^2+^ represents the increase in oxygen vacancy content. On the one hand, the increase in Co^2+^ will accelerate the formation of the intermediate product CoOOH, thereby improving the catalytic activity of oxygen evolution [34]. On the other hand, the increase in Co^2+^ content stems from the generation of more surface oxygen vacancies. The absence of oxygen will delocalize the two electrons that previously occupied the O 2p orbital around the nearby Co^3+^ to form low-coordination Co^3+^, which will facilitate the adsorption of reactants. The delocalized electrons are easily excited to the conduction bond to improve the conductivity of the material. The O 1 s XPS spectra were further analyzed to explore the oxygen species on the catalyst surface. The fitted O 1s spectra (Figure 3e,f) contain three oxygen species, O_I_ peak at ~529.5 eV, O_II_ peak at ~532.3 eV, and O_III_ peak at ~531.1 eV, corresponding to the metal–oxygen bond, oxygen vacancy, and the hydroxyl group adsorbed by oxygen onto the surface. The ratio of O_II_ in Co/Ce0.2-COHDs is 32.1%, which is higher than the ratio in Co_3_O_4_-HDs (29.3%). This result further illustrates that the introduction of CeO_2_ phase can increase the oxygen vacancy content in the material.

In order to further illustrate the importance of CeO_2_, a well-defined CeO_2_(111)/Co3O_4_(220) nanointerface (Figure 4a) was constructed based on density functional theory (DFT) theory. Figure 4a shows the crystal structure of Co/Ce0.2-COHDs. For the spinel Co_3_O_4_ phase, Co^3+^ ions occupy half of the octahedral position, and Co^2+^ ions occupy 1/8 of the tetrahedral position. The mixed valence of Co cations provides donor–acceptor chemisorption sites for oxygen adsorption, making Co_3_O_4_ a promising candidate for OER [35]. The unit cell of CeO_2_ is a typical face-centered cubic fluorite structure. The coexistence of Ce^3+^ and Ce^4+^ provides good surface oxygen ion exchange performance and good electrical conductivity. Therefore, the combination of CeO_2_ and Co_3_O_4_ will produce a synergistic coupling effect, which is expected to improve its electrocatalytic oxygen evolution performance. As shown in Figure 4b, some electron coupling effects were observed at the interface between CeO_2_ and Co_3_O_4_. The obvious electron transfer from CeO_2_ to Co_3_O_4_ at the interface is clearly observed, which will contribute to modulating the electron density of the Co sites (Figure 4b). This calculation result is basically consistent with the experiment. The occurrence of electron transfer will lead to the generation of low-coordination Co. At the two-phase interface of Co_3_O_4_ and CeO_2_, equivalent negative and positive charges will appear due to electron coupling (Figure 4c), thus forming a channel for fast electron transfer. Furthermore, Appendix A shows that the density of states (DOS) of the Co/Ce0.2-COHDs interface at the Fermi level is not zero, illustrating the enhanced conductivity and reflecting the metallic nature of the interface [30,36,37]. Theoretical analysis also demonstrates that the gradient orbital coupling reinforces the Co-O covalency of the Ce(4f)-O(2p)-Co(3d) unit active site with an optimized Co-3d-eg occupancy, which can balance the adsorption strength of intermediates and in turn reach the apex of the theoretical OER maximum [38].

The electrocatalytic OER activity of the as-prepared Co/Cex-COHDs was evaluated in a 1.0 M KOH electrolyte with a three-electrode system. Figure 5a and Appendix A depict the linear sweep voltammetry (LSV) curve of as-prepared series of Co/Cex-COHDs. From the trend of the LSV curve (Figure 5a), it was observed that the OER activity of Co_3_O_4_-HDs that did not form a composite structure with CeO_2_ was poor. After CeO_2_ is anchored to Co_3_O_4_, the OER activity of the Co/Cex-COHDs was significantly increased compared with Co_3_O_4_-HDs (Figure 5a,b). When the Ce ratio reaches 0.2, Co/Ce0.2-COHDs achieves the optimal OER catalytic activity with an overpotential of only 316 mV at a current density of 10 mA·cm^−2^ (Figure 5a,b). Such a small overpotential represents lower energy consumption during oxygen evolution. However, the catalytic activity of Co/Ce0.3-COHDs has become very poor (Figure 5a), and its overpotential at 10 mA·cm^−2^ is up to 482 mV. Only the overpotential at a certain current density is not enough to evaluate the merits of a catalyst because it may not fit the trend of overpotential at a high current density. Promisingly, the Tafel slope derived from the LSV curve, which represents how fast the overpotential changes with current density, will solve this issue. Figure 5c shows the Tafel plots of Co/Cex-COHDs. It was observed that Co/Ce0.2-COHDs had the lowest Tafel slope (89.9 mV·dec^−1^) compared to other catalysts, which indicates a better kinetic process. Subsequently, the turnover frequency (TOF) was calculated to investigate the intrinsic catalytic activity of the catalyst. As shown in Figure 5d and Appendix A, the TOF value of Co/Ce0.2-COHDs is as high as 0.16 s^−1^, which is 1.3 times that of Co/Ce0.1-COHDs (0.12 s^−1^), 53 times that of Co/Ce0.3-COHDs (0.003 s^−1^), and 9.4 times that of Co_3_O_4_-HDs (0.017 s^−1^). A higher TOF value of Co/Ce0.2-COHDs means that more oxygen can be produced per unit of time. In addition, mass activity is a key indicator for evaluating electrocatalytic performance. As indicated in Figure 5e and Appendix A, Co/Ce0.2-COHDs achieve mass activity up to 18.7 A·g^−1^ at a given overpotential of 350 mV, 1.4 times higher than Co/Ce0.1-COHDs, 57 times higher than Co/Ce0.3-COHDs, and 10 times higher than Co_3_O_4_-HDs. To account for the difference in performance, the electrochemical surface area (ECSA) was tested and calculated by monitoring the current density in the non-Faraday zone with different scan rates (Appendix A). According to the formula ECSA = C_dl_/C_s_ (C_dl_ is double layer capacitance, C_s_ is specific capacitance), ECSA is proportional to C_dl_. As displayed in Figure 5f, Co_3_O_4_-HDs exhibits a small slope, that is, a small C_dl_. With the introduction and increase in CeO_2_, the C_dl_ of the Co/Cex-COHDs gradually increases and reaches a maximum at a Ce ratio of 0.2. The larger C_dl_ implies a larger ECSA, which reflects the existence of more-effective active sites. Continuing to increase CeO_2_ leads to a severe decrease in the C_dl_ of the material, mainly due to a sharp decrease in the number of active sites. Moreover, the effect of calcination temperature on the OER performance of the sample was further discussed. It can be seen from the LSV curves and Tafel plots (Appendix A) that lower and higher temperature will lead to lower OER performance. On the one hand, lower temperature will lead to excessive oxygen vacancies (Appendix A), thus reducing the conductivity of the material [39]. On the other hand, higher temperature will increase the grain size, which will reduce the number of exposed active sites. To clarify the role of hollow structures, Co/Ce0.2-CONPs are synthesized (Appendix A). Compared with Co/Ce0.2-CONPs, Co/Ce0.2-COHDs exhibit better OER performance (Appendix A), which is mainly because the unique hollow dodecahedron structure assembled by nanoparticles not only can effectively avoid the grain accumulation but also provides plenty of channels for the diffusion of electrolyte and gas transportation, thus accelerating the reaction oxygen evolution kinetic process. Finally, a chronopotentiometry test was used to explore the durability of Co/Ce0.2-COHDs during the OER process. The test result shows that Co/Ce0.2-COHDs can maintain stability for more than 10 h with almost no degeneration (Appendix A). After that, the morphology and structural stability of Co/Ce0.2-COHDs were studied. As shown in Appendix A, the morphology and phase of Co/Ce0.2-COHDs remain basically unchanged after OER. It can be seen from Appendix A that Co/Ce0.2-COHDs also exhibit better performance than similar catalysts reported in the literature [40,41,42,43,44,45,46,47,48,49].

## 3. Experimental Setup

Cobalt nitrate hexahydrate (Co(NO_3_)_2_·6H_2_O), cerium nitrate hexahydrate (Ce(NO_3_)_3_·6H_2_O), 2-methylimidazole (C_4_H_6_N_2_), methanol (CH_3_OH), potassium hydroxide (KOH), deionized water (H_2_O), and Nafion (5%) were purchased from Shanghai Aladdin Biochemical Technology Co., Ltd. (Shanghai, China). All reagents for the synthesis of catalysts were of analytical grade and were used without further purification.

First, cobalt nitrate hexahydrate and cerium nitrate hexahydrate were dissolved in 20 mL methanol (Solution A) according to the stoichiometry in Appendix A. Meanwhile, 0.656 g 2-methylimidazole was added to 100 mL methanol and the mixture was stirred continuously for 5 min to form a clear solution (Solution B). Then, a blue precipitate formed rapidly after the slow addition of solution A to solution B, followed by continuous stirring for 5 min. Subsequently, the suspension was aged for 24 h and a blue precipitate was obtained by centrifugation. The precipitate was washed three times with water and ethanol and then dried in an oven at 80 °C. Finally, Co/Ce-MOFs with different Ce ratios were successfully obtained. The pre-prepared Co/Ce-MOFs were heated to 350 °C at a heating rate of 1 °C·min^−1^ and maintained for 3 h. Finally, Co/Cex-COHDs with Ce ratios (x = 0, 0.1, 0.2, and 0.3) were successfully prepared and named Co_3_O_4_-HDs, Co/Ce0.1-COHDs, Co/Ce0.2-COHDs, and Co/Ce0.3-COHDs, respectively. Furthermore, the Co/Ce0.2-COHDs at different calcination temperatures (300 °C, 400 °C) were obtained by adjusting the calcination temperature of the pre-prepared Co/Ce-MOFs with a Ce ratio of 0.2 and were named Co/Ce0.2-300-COHDs and Co/Ce0.2-400-COHDs, respectively. Next, 466 mg Co(NO_3_)_2_·6H_2_O and 147 mg Ce(NO_3_)_3_·6H_2_O were dissolved in deionized water to form a clear solution. The prepared excess 0.5 M KOH solution was added to the above solution and stirred continuously for 10 min. Then, the suspension was centrifuged and the precipitate was washed with water and ethanol. Subsequently, the precipitate was dried in a 60 °C oven. Finally, the sample was calcined in a tubular furnace; the heating rate was 1 °C·min^−1^, the holding temperature was set at 300 °C, and the holding time was 3 h. After the above preparation process, Co/Ce0.2 composite oxide nanoparticles (Co/Ce0.2-CONPs) were finally prepared.

XRD patterns of all the samples were collected in the range from 30° to 70° using a BRUKER D8 ADVANCE X-ray powder diffractometer (Cu K radiation, λ = 1.5406 Å), operated at 40 kV and 100 mA. The specific surface area and pore size distribution are usually determined by N_2_ adsorption at 77 K using the BET and BJH methods on SSA-7000 sorption analyzer. The morphologies of all the samples were characterized by scanning electron microscope (SEM, JSM-6700) and transmission electron microscope (TEM, JEM-2100). HAADF-STEM image and elemental mapping were collected using a TEM (JEM-2100F) equipped with EDX spectroscopy. The XPS spectra were collected by a Thermo K-Alpha+ X-ray photoelectron spectrometer.

The electrochemical measurements were carried out with a typical three-electrode system at room temperature. First, 5 mg catalyst was dispersed in 500 μL ethanol. Then, 30 μL Nafion solution (5%) was added to it. The above mixture was sonicated to form a homogeneous ink. Finally, the experimentally optimized 20 μL mixed ink was dispersed onto a glassy carbon electrode with a diameter of 5 mm and dried naturally. The electrochemical tests were conducted in an alkaline solution (1.0 M KOH) using a standard three-electrode system at room temperature on a CHI660E electrochemical workstation. Catalysts cast on glassy carbon electrode was used as the working electrode. A carbon rod was used as the counter electrode. Ag/AgCl (saturated KCl) was used as the reference electrode. Polarization curves were acquired by linear sweep voltammetry (LSV) that swept from 0 to 0.8 V versus Ag/AgCl at a scan rate of 10 mV·s^−1^. The LSV curves were plotted as overpotential (η) versus the logarithm of current density (log|j|) to obtain Tafel plots. The electrochemical double-layer capacitance (C_dl_) was performed through the cyclic voltammetry (CV) method, and the scan rates of potential were set as 10, 20, 30, 40, and 50 mV·s^−1^ from 0.1 to 0.2 V versus Ag/AgCl, where no faradic current was observed.

The first-principles calculations were performed by using the Vienna ab initio simulation packages3 (VASP). The calculation parameters are as follows. The Perdew–Burke–Ernzerhof (PBE) parametrization of the generalized gradient approximation is for the exchange–correlation function. The energy cut-off of the plane wave basis was set as 500 eV and a gamma centered 5 × 5 × 1 Monkhorst–Pack k-point mesh was applied for the k-point samples in the Brillouin zone. A vacuum space of about 20 Å was used to avoid spurious interactions. The tolerance of electronic and ionic relaxation was 10^−5^ eV and 0.01 eV·A^−1^, respectively. Based on later experimental characterization and observation, the Co_3_O_4_/CeO_2_ model was employed from CeO_2_(111) plane and Co_3_O_4_(220) plane. To simulate the interface structure of Co/Cex-COHDs, the structures of CeO_2_ and Co_3_O_4_ were firstly calculated. The Co_3_O_4_ has cubic Fm-3m symmetry and the lattice constant a is 8.012 Å. CeO_2_ also has cubic Fm-3m symmetry and lattice constant is 5.402 Å.

Turnover frequency (TOF) represents the number of O_2_ generated per unit of time at a single active site. The TOF value can be calculated according to the following formula [50]. Since the Ce site is almost inactive in these Co/Cex-COHDs, the Co site is assumed to be the active site in this calculation [51,52].
$$TOF(\times 10^{-2}\;s^{-1})=\frac{j(mA\cdot cm^{-2})\times M(g\cdot mol^{-1})}{F(C\cdot mol^{-1})\times n\times m(mg\cdot cm^{-2})}$$
where j is the current density at a given overpotential of 350 mV, *M* is the molar mass of the catalyst, *F* is the Faraday constant (*F* = 96,485 C·mol^−1^), *n* is the number of electrons transferred during the oxygen evolution reaction (*n* = 4), and *m* is the loading of catalyst on the electrode.

## 4. Conclusions

In summary, the Co/Cex-CONPs with different Ce ratios were successfully fabricated by a hard-template strategy using MOFs as self-sacrifice template, ensuring the controllability of catalyst morphology and the uniformity of the composition. The optimized Co/Ce0.2-COHDs achieve the best OER performance, i.e., the overpotential at 10 mA·cm^−2^ is only 316 mV and the Tafel slope is 89.9 mV·dec^−1^, which is lower than the others. The introduction of an appropriate amount of CeO_2_ can effectively improve the electron distribution, accelerate the charge transport, and optimize the adsorption of reaction intermediates. However, excess CeO_2_ will cause the number of active sites to decrease and the microstructure to collapse, resulting in poor catalytic activity. More critically, the construction of a hollow dodecahedral microstructure can provide a larger effective surface area to expose more active sites and a rich pore structure to accelerate the diffusion of ions and generated gases.

## Data Availability

All data are available in the main text or the Appendix A.

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
