# Peer review of "Oxygen Vacancy and Interface Effect Adjusted Hollow Dodecahedrons for Efficient Oxygen Evolution Reaction"

_molecules, 2023, doi:10.3390/molecules28155620_

Round 1

Reviewer 1 Report

In this paper, the authors describe a work about preparing of Co/Cex-COHDs for electrocatalytic application. The current manuscript need minor revisions before acceptance and some recommendations are listed.

 1. What is the benefit of your composites compared with other Co/Ce composites structures?

2. It is better to provide a scheme for the experiment. Separate the characterization and synthesis process with different titles.

Reviewer 2 Report

Review (minor revision):

In the manuscript titled; “Oxygen vacancy and interface effect engineering dual modulation of cobalt-cerium composite oxide hollow dodecahedrons for boosting oxygen evolution reaction” by the authors' Wang et al., the authors studied the oxygen evolution reaction using electro-catalyst based Co and Ce MOFs.

Introduction:

The introduction part is well-written and clear.

Experimental Setup:

The authors should correct the statement: “Brunauer-Emmett-Teller (BET) surface area…”. One does not measure BET surface area, you measure the amount of sorbed nitrogen. The specific surface area and pore size distribution are usually determined by applying the BET and BJH methods.

In the section describing XPS, the authors should define what standard was used to offset sample charging. Did the authors perform symmetrical or asymmetrical peak fitting? What was the reduced chi squared?

Results and Discussion:

The elemental mapping shown in Figure S1 is barely visible. The authors should improve the contrast between the elements and the black background.

In the section describing XRD, the authors should elaborate on how the grain size was determined. Calculated using the Scherrer equation or measured from TEM.

In Figure 2d Curve iv, it can be clearly seen that the diffractions attributed to the Co3O4 phase completely disappear. On the other hand, the peaks associated with the Ce phase seem unchanged. What is the authors’ explanation for this phenomenon?

In the XPS results of Ce, the authors omit the peaks at approximately 882.5 and 901 eV. These peaks are also associated with the Ce4+ 3d orbital (10.1002/sia.740200604).

The peak deconvolution for Co2p is not quite accurate. According to Biesinger et al (10.1016/j.apsusc.2010.10.051), determining the amount of Co+ is not as straightforward as presented by the authors. Looking at the graphs, I would assume that in both cases Co3O4 is the dominant phase and that there is negligible shift in the ratio of both oxidation states of Co.

The authors should take care in calculating TOFs. Since the authors assume Co's grain/nanoparticle size to be about 25 nm, only about 15 % of atoms are on the surface of such particles.

Conclusion:

The conclusion part is well written.

Minimal improvements in English are necessary.

Reviewer 3 Report

Wang et al. present their work on CeO2/Co3O4 composites using both experimental and ab initio approaches. The obtained composites have structural and electronic advantages as a catalyst for water splitting. The topic is rather interesting as nowadays H2 production from water has been an important topic. The experiments were described in detail. The samples were characterized well. The code VASP has been well-established and the PBE-GGA approximation is suitable for the present study. The settings seem OK. The frame of the manuscript is proper. The text is in the scope of this Journal. Thus, I’d like to suggest acceptance of this manuscript for publication in Molecules after improvements.

1). The title is too long and complex. It can be shortened to contain the essence of this work.

2). Lines 41 to 44 (citation here) ‘CeO2, with strong ability to shift between Ce3+ and Ce4+ derived from the unique 4f 5d electronic configuration of Ce, exhibits excellent properties including reversible surface oxygen’ (end of citation) is inconsistent and can be misleading. Indeed, Ce can exhibit 3+ or 4+ in the oxidation however, CeO2 itself is Ce4+, whereas in Ce2O3 is there C3+ according to the ionic model.

3). Line 123 ‘Perdew-Burke-Ernzerhof (PBE)4 parametrization’ seems strange. What is meaning of the (PBE)4 ?

4). There is a lack of details about the CeO2/Co3O4 interface system (lines 126-129). It is vital to provide details about the cells and atomic arrangements, which helps readers to understand the reliability and physical meaning of the obtained results. Also please give reasons for the usage of the orientation relation for the interface system in line 129.

5). It might be useful for readers if the authors give some description about the crystal structures of CeO2 and Co3O4. Moreover, please give information about the lattice matching between CeO2(111) and Co3O4(110) ?  It is well-known that misfits in different orientations may have strong impacts on the optimized local structure and related electronic properties.

6). The 3d orbitals in cobalt oxides are localized. Thus, it is necessary to employ some advanced approaches to deal with it. One cheap approach is Hubbard U approximation. I’d like to suggest the authors spend time to recalculate the interface system using the GGA+U approach. Moreover, the role of Ce 4f electron in Ce3+ at the interface should be addressed.

7). The formula to calculate Turnover Frequency (TOF) is created by the authors themselves or from literature? If the latter, it is necessary to give the references.

8). From Figure 2, I could not recognize the ORs between the two oxides crystallites. It would be nice if the authors make an enlarge TEM image and schematic atomic arrangements at the interfaces, which may support the interface model for first-principles simulations.

9). Magnetism is important in water splitting since O2 molecules are magnetic. I’d like to suggest the authors addressing this point in the introduction and discussions.  

10). Figure 4 provided readers some impression about the interface charge transfer. However, a more direct approach, such as Bader charge model may be employed to produce values of charges at the atomic sites around the interfaces.

There is some space to improve the language and descriptions.

Round 2

Reviewer 3 Report

I am OK with the responses and improvements of the manuscript.